# Selecting Nomination Committee Members—Stakeholders' Perspective

**Hildur Magnusdottir [1], Audur Arna Arnardottir [2] and Throstur Olaf Sigurjonsson [1,3,*]**

[1] Institute for Experimental Pathology, University of Iceland, 101 Reykjavik, Iceland
[2] School of Business, Reykjavik University, 101 Reykjavik, Iceland
[3] Department of International Economics and Management, Copenhagen Business School, 2000 Frederiksberg, Denmark
[*] Correspondence: olaf@hi.is; Tel.: +354-5454000

**Abstract:** Sustainability is a critical issue for businesses today, and corporate boards and nomination committees play a vital role in promoting sustainable practices within organizations. Nomination Committees (NCs) have become important mechanisms for what has been coined as good or sustainable governance. Stakeholders, however, have different opinions on the merits of these committees. Amongst other things, there is disagreement about whom to select for NCs and the criteria for selection. The composition of NCs can influence board nominations and determine whether the necessary knowledge and skills reside within a board to deal with sustainability and good corporate governance. The aim of this research is to provide insights into stakeholders' perspectives on what matters when profiling nomination committees' members. The literature on NCs is underdeveloped, and this research importantly addresses the issue by interviewing 13 individuals who are either shareholders, board members of listed companies, or members of NCs. Additionally, a questionnaire was sent to shareholders, board members, and NC members of the 300 largest companies in Iceland. A total of 138 responses were received. The results show that stakeholders believe it is important to decide upon an NC's composition prior to electing its members. NC members should have either management or board membership experience. Diversity within the committees is also seen to be important, as it can influence diversity and inclusion overall at the board level. These research results provide important insights and understanding for shareholders and boards of directors when composing nomination committees.

**Keywords:** nomination committees; corporate governance; sustainable governance; board committee; committee members

## 1. Introduction

Incorporating sustainability considerations into the governance of a company requires a comprehensive approach that encompasses environmental, social, and governance (ESG) factors. Corporate governance codes encourage companies to establish Nomination Committees (NCs) to strengthen the process of identifying and selecting board members and thus foster sustainable or good governance [1]. However, there are no legislative requirements for companies to establish such committees [2]. Instructions for how to operate NCs have been part of corporate governance guidelines in all of the Nordic countries, including Iceland [3–6]. In the Nordics, NCs are considered advisory committees that propose board members for shareholder votes at Annual General Meetings (AGMs). NCs, along with Audit and Remuneration Committees, are the three kinds of committees commonly established within corporate governance structures.

The NC's role is twofold. First, the committee must establish criteria for necessary skills that directors of a board must hold. The second is to approach and evaluate potential board candidates [7–9]. Previous research has explored the purpose of NCs, but it is not always clear whether the committees are only formed to comply with rules and guidelines

or whether they are established to improve the process [2,9]. NCs have, however, according to Ruigrok et al. [9], changed the process of how board members are selected.

Before NCs were introduced, shareholders were often in control of nominating and electing board members [1]. A CEO and/or a board chairperson also sometimes played a role when selecting new board members [2,7]. Researchers have argued that when shareholders nominate board members, the board's work is possibly affected, especially if shareholders do not have sufficient information or knowledge needed to appoint qualified directors [10]. In Sweden, for example, before NCs were established, the nomination process involved a leading owner and the chair gathering a group of the largest shareholders, with a combined majority of voting rights, to discuss a renewal of the board [2].

As NCs are expected to nominate the most qualified individuals for companies' boards of directors, this brings attention to the composition of NC committees themselves. Limited research on the profiles of NC members has been conducted. Studies have focused on the role of the committee in relation to the importance of the diversity of committee members [9,11]. Research has shown the importance of diversity to NCs, so that the final board of directors ends up being diverse as well [9,11]. Research confirms that the professional experience of NC members is important, as is having a strong 'network'. At the same time, NCs need to be familiar with a company's operations [2]. However, there is a lack of pertinent research on what qualities and experience need to be considered when choosing NC members. Current research on NC members is limited, as is research on NCs members' qualifications, education, and knowledge [12]. In Iceland, listed companies rely partly on corporate governance guidelines published by the Iceland Chamber of Commerce when making decisions about NC selections. The corporate governance guidelines do not specify the specific qualifications of NC members [6]. They do, however, state that most NC members must be independent of the relevant company and its management and that at least one member should be independent of the largest shareholders [6]. Iceland has been considered to have a similar corporate governance structure to other Nordic countries, namely, Norway, Sweden, and Denmark. The Nordic Corporate Governance model has been considered to be similar to a Two-Tier model that has been used in Germany, Denmark, Norway, and Sweden [10,13]. According to Lekvall [13], the Nordic Model is used in Nordic countries and resembles the Two-Tier structure. Therefore, exploring NCs in Iceland can be relevant for other Nordic countries and countries adhering to the Two-Tier structure.

This research applies data received from shareholders, board members, and NC members in order to expand the literature on what knowledge and experience NC members should possess. The aim of this research is to understand what qualifications various stakeholders see as necessary when considering whom to choose for an NC. This article is structured by first presenting a theoretical overview along with a comparison of how NCs are organized within Icelandic Corporate Governance Guidelines and guidelines of other Nordic countries. Next, the research methodology is presented, followed by the results, discussion, and closing remarks.

## 2. Literature Review

### 2.1. Nomination Committee Members

Having an independent NC can determine the profiles of the directors who will possibly be assigned to the board [8]. However, there are different views on who should be appointed to an NC. Ruigrok et al. [9] have found that it is important to have an NC in the first place, i.e., an NC can increase the likelihood of selecting board directors who are more likely to protect the interests of all shareholders [8].

Committee composition can also be important. When there is continuity in the composition and if the ownership structure and representation of the NC are relatively stable, the NC functions better. The main advantage of having an NC is that companies can benefit from the experience of those who have previously participated in a recruitment process. It can also promote the engagement of its owners [2].

The NC's influence is also based in part on what total capital the NC represents or what voting rights or shareholdings the NC members represent [2]. According to Eminet and Guedri [8], the CEO and other executive directors should not have seats on the NC, as it is then considered better equipped to resist the influence that the CEO might have on the selection process for new directors. If the NC does not include the CEO and outside directors dominate the committee, it can decrease the CEO's influence. Research on the 200 largest companies in France between 2001 and 2004 showed that the profiles of directors who are likely to be appointed as board members are determined by the existence and independence of NCs [8].

The diversity of NC members affects the diversity of the board of directors they nominate [10,11]. If an NC has a diverse race and gender composition, it is more likely to appoint a diverse board and to determine whether the necessary knowledge and skills reside within a board to deal with sustainability and good corporate governance [11]. Ruigrok et al. [10] also found that when NCs exist, boards are more likely to have more foreign directors as well as higher degrees of national diversity. Other research suggests that when a foreign person has a seat on an NC, this increases the likelihood of more national diversity within the board of directors [14]. Hutchinson et al. [15] found that if females are on an NC, more women are appointed to the board. Kaczmarek et al. [14] argue that, in the UK, gender-diverse NCs are more likely to propose female candidates to the board. Simply put, if an NC has diverse members, it is more likely to select from a diverse pool of candidates. At the same time, having an NC does not necessarily guarantee the appropriate education and gender diversity of the board [9].

NC members' standing and status are also significant, as are their network and competence. A committee member's network is important, as knowledge of and access to a diverse range of candidates is essential in recruitment situations [2]. To multiply the NC's network, they can use headhunters, employment agencies, or consultants [2,11]. These resources can, in addition to providing a larger network [11], also assist in facilitating the committee's work. It can also be useful for NC members to have specialist competence or experience in the company's operations or business context. The social competencies of individual members of the committee, their experience, and know-how are also factors that can be beneficial for NCs. Some experts purport that NC members' key competencies should include familiarity with the operations and strategy of the company so that they can better understand dependencies, risks, and the most relevant circumstances [2].

*2.2. Nomination Committee Members in the Nordic Countries and Corporate Governance Models*

According to the OECD [16], different models of good corporate governance exist globally. The general corporate governance principles published by the OECD provide a flexible and robust reference for market participants and policymakers to create their own corporate governance frameworks. They further state there is no single model of corporate governance and that corporate governance regulations and rules should be adapted to the reality within which they are implemented [16]. Different corporate governance structures have been defined as One- and Two-Tier structures and are often referred to as board systems [2,10,13,17]. Lekvall [13] also defines a third structure: the Nordic Model. The One-Tier model is used in the US [10,13], the UK, China, and Japan [10], while the Two-Tier system has been used in Germany, Denmark, Norway, and Sweden [10]. According to Lekvall [13], the Nordic Model is used in the Nordic countries and resembles the Two-Tier structure. Companies in some countries are able to choose which structure they use, such as Denmark [10] and France [10,18]. Most companies in Denmark choose the Two-Tier model, while the One-Tier model is more popular in France [10]. The three models all have the general (shareholders') meeting at the top of the ownership level in order to elect the board [10,13].

The Two-Tier model is implemented where country-specific law requires companies to have two management levels [10]. Under this model, a supervisory board is elected by the shareholders and should have oversight and control over the management board [10,13].

The Nordic countries are considered by Lekvall to have country-specific variations [13]. Nevertheless, Sjöstrand et al. [2] report that the similarities outweigh the differences. There are some similarities within the Nordic countries regarding regulations, both soft and hard. According to Sjöstrand et al. [2], the Nordic model gives companies the flexibility to create their own solutions, as corporate governance can be tailored to suit each company. This flexibility is considered one of the main strengths of the Nordic model; companies do not have to comply with standardized solutions regarding owner structures, categories, or forms.

When considering the Corporate Governance Guidelines of Nordic Countries, they have similar rules regarding NC members' independence, i.e., most members should be independent of the company [3–5,15]. The Swedish guidelines also state that at least one member of the committee should be independent of the largest shareholder [5]. In Norway, the NC should be independent of the board and executive personnel [4].

According to the corporate governance rules in Iceland, NCs are to be composed of at least three members. However, they may consist only of two if both members are independent of major shareholders [6]. Swedish and Finnish NCs should also have at least three members [5,14]. There are no requirements on how many members should be appointed to NCs in Denmark and Norway [3,4]. Board members can be members of NCs in the Nordic countries, though they are subject to few conditions in Iceland, Norway, and Sweden [3–5,19]. If board members, whether the chair of the board or another director, are members of an Icelandic NC, they cannot chair the committee.

The guidelines also stipulate that NC members "may not constitute a majority of the Committee" [6] (p. 14). According to this, the NC should not be entirely made up of board members [6]. The Swedish guidelines have similar provisions, as board members cannot hold a majority in the NC [5]. In Norway, only one board member can participate, and only if they are not considering re-election [4]. The reasoning for having board members on a committee is, according to the Icelandic corporate governance guidelines, that it can provide the NC with a certain overview of a board's work. Furthermore, the company's employees or its managers should not be members of the NC [6].

Considering the above, many questions remain regarding NCs and what knowledge, experience, and qualities NC members should possess. In order to expand the literature on NC membership, we aim to answer the research question: What knowledge, experience, and qualities are deemed to be important for NC members according to different groups of stakeholders?

## 3. Research Design and Methods

It was considered beneficial to collect both qualitative and quantitative data for this research to gain more insight into Icelandic NCs, their role, and who should be appointed as NC members. The qualitative data were collected through interviews with stakeholders. Quantitative data were gathered from an online survey to gain further insight into how principal stakeholders feel about the necessary qualities of NC members. The qualitative interview guide was constructed after the researchers attended a meeting held by the Iceland Chamber of Commerce, where NCs were discussed by investors, board members, and NC members. The questionnaire Clune et al. [20] used in their research was employed, as were previous studies on NCs (see Appendix A). The quantitative questionnaire was constructed after the qualitative data had been gathered. It was constructed based on the same resource as the semi-structured interview guide. However, as the interviews had already been conducted, the information gathered during the interviews affected the questionnaire. Therefore, the questionnaire was designed to answer lingering questions to gain further insight into how a larger sample viewed NC committees, how they should be structured, and the selection of NC members.

The results of both research approaches were then compared to examine the coherence of the data in relation to the research question, and this was supported by descriptive

statistics and thematic analysis. This chapter will cover the research methods used and how pertinent data were collected and processed.

### 3.1. Qualitative Methodology

Qualitative data were collected by conducting thirteen semi-structured interviews. The semi-structured interview frame was partly based on a study by Clune et al. [20]. Other questions were constructed based on previous discussions with various stakeholders attending an event for stakeholders of listed companies hosted by the Iceland Chamber of Commerce.

The semi-structured interview guide was divided into three parts (see Appendix A). The first part included background questions for participants. The second part included questions on the role and structure of NCs and the pros and cons of having an NC. The third part included questions on the work processes of NCs. Two versions of the guide were constructed, one for NC members and one for shareholders and board members.

Each interview lasted between 30 and 90 min; the average length was 63 min. The interviews were conducted face-to-face, except for two conducted online due to the COVID-19 pandemic. The participants were informed about the research and the purpose of the interviews. Full confidentiality and anonymity were assured; all interviewees (with one exception) permitted recording of the interviews and agreed to be quoted. Notes were taken during the interview that was not recorded. To achieve the best possible understanding of the different viewpoints, NC members, board members of registered companies, and investors were interviewed. Interviewees were selected based on their experience with NCs and were either shareholders, NC members, or board members. The interviews were conducted in February and March 2020, except for two interviews that were conducted via teleconferencing equipment in June 2020. The interviewees were contacted by e-mail, where the aim of the study was presented. A descriptive list of participants is found in Table 1.

**Table 1.** Interviewees.

| Participant | Gender | Role |
|---|---|---|
| 1.N | Female | NC member |
| 2.N | Female | NC member |
| 3.N | Female | NC member |
| 4.N | Female | NC member |
| 5.N | Male | NC member |
| 6.N | Female | NC member |
| 7.N | Female | NC member, board member, consultant |
| 8.N-B | Male | NC member, board member |
| 9.N-B | Male | NC member, board member |
| 10.B | Male | Board member |
| 11.I | Female | Institutional investor |
| 12.I | Male | Institutional investor |
| 13.C | Male | Consultant |

The participants were given a number and a reference to his/her role, e.g., participant one, an NC member, had the number "1.N". Participants who were both NC members and board members of the same company were referenced with "N-B", e.g., participant number eight was referred to as "8.N-B". Investors were given an "I", while the consultant is marked with a "C".

After the research data were collected, the interviews were transcribed. Grounded theory was used to systematically analyze the data [21]. The transcribed interviews were coded, with coding used to assign shorthand designations to different aspects of the data so that the researcher could quickly access specific data items. Open coding was used, which means that important points or phrases were marked for further data analysis [22]. The data were coded using the program NVivo, where themes and subcategories were

defined according to the nature of the questions and the categorization of the interviewee's main motivations. All interviews were coded by one researcher and reviewed by two co-researchers to ensure the internal validity of the coding. The themes derived from the coding of the interviews can be found in Table 2, below.

**Table 2.** NVivo codes and themes.

| Topic | Participants | Cases |
|---|---|---|
| **Nomination Committees in Iceland** | **13** | **105** |
| Establishment of NCs in Iceland | 8 | 13 |
| Need for NCs in Iceland | 12 | 92 |
| **Shareholders' Committee or Board Committee** | **12** | **49** |
| Voted by shareholders | 9 | 22 |
| Independent committee | 9 | 11 |
| Logical structure | 7 | 16 |
| **NC Members** | **13** | **88** |
| Knowledge and experience | 13 | 50 |
| Other factors when choosing NC members | 13 | 38 |
| **Board Members as NC Members** | **13** | **107** |
| Connection with the board | 6 | 10 |
| Gathering information | 10 | 25 |
| Restrictions regarding board members | 11 | 47 |
| Responsibility of other NC members | 8 | 25 |
| **Future of Icelandic NCs** | **12** | **110** |
| Attitude towards NCs | 10 | 29 |
| Development of NCs in Iceland | 11 | 81 |

*3.2. Quantitative Methodology*

In cooperation with the Iceland Chamber of Commerce, a Quantitative Questionnaire was sent to 542 people, all shareholders in listed companies, NC members, board members of listed firms, or representatives of the 300 largest Icelandic companies. The total number of participants was 138, and the response rate was 25%. The pool of participants represented most shareholders, NC members, and board members of the listed companies in Iceland, as the questionnaire was sent out to all relevant parties. The majority of the participants' email addresses were registered with the Iceland Chamber of Commerce, but, additionally, the authors gathered emails to ensure better participation.

The questionnaire was divided into four parts (see Appendix B). The first part included background questions regarding age, gender, education, and whether the participants were NC members, board members, and/or shareholders. The next part contained questions from the Iceland Chamber of Commerce about guidelines on governance. The third part explored the Nomination Committees themselves, soliciting answers to questions such as: how they should be composed, who should sit on the committees, and how satisfied were the participants with the NC committees. The last part was only for current or previous NC members.

The survey participants were asked to state on a 5-point Likert scale how strongly they agreed or disagreed with five statements regarding what experience at least one member of an NC should have. The statements that followed were knowledge of recruitment, legal knowledge, corporate governance knowledge, experience as a board member, and managerial experience. The participants were also asked to rank which of the listed twelve qualities are important for NC members. They were asked to rank them on a spectrum from 1 to 10, where 1 was very insignificant and 10 was very important.

The participants were divided into four groups, as can be seen in Table 3, below. The first group comprised shareholders, both institutional and private investors. The next group included board members of listed companies, and the third group contained NC members, both those who were on committees when the survey was conducted and those who had previously been on such committees. The fourth and last group included those on the boards of unlisted companies, who were not NC members or shareholders.

**Table 3.** Survey participants: stakeholder groups.

| Role | Quantity | % |
| --- | --- | --- |
| NC members | 20 | 14% |
| Board members of listed companies | 16 | 12% |
| Shareholders | 62 | 45% |
| Others | 40 | 29% |

Of the 138 participants, 66% were male and 63% of the participants were 50 years of age or older, as shown in Table 4.

**Table 4.** Survey participants: age and gender.

| Participants | Variable | Quantity | % |
| --- | --- | --- | --- |
| | Female | 46 | 33% |
| Gender | Male | 91 | 66% |
| | Other/no answer | 1 | 1% |
| | 30–39 years | 16 | 12% |
| | 40–49 years | 36 | 26% |
| Age | 50–59 years | 55 | 40% |
| | 60 years or older | 31 | 23% |

## 4. Results

This section describes the results from the interviews and the survey that was conducted, focusing first on the knowledge and experience of NC members and then on the qualities and skills of members, according to both the interviewees and the survey participants. The findings shed light on desirable qualifications, education, and skills for NC members.

### 4.1. Knowledge and Experience of NCs

When participants were asked to decide how important it is for NC members to have different types of knowledge and experience, the findings show that it was most important for NC members to have managerial experience (M = 4.46, SD = 0.67) when compared to experience as a board member, legal knowledge, corporate governance knowledge, and recruitment knowledge, as shown in Table 5 below.

**Table 5.** Survey participants.

| | Mean | Standard Deviation |
| --- | --- | --- |
| Knowledge in Recruitment | 3.64 | 1.04 |
| Legal Knowledge | 3.01 | 0.95 |
| Corporate Governance Knowledge | 4.31 | 0.64 |
| Experience as Board Member | 4.27 | 0.73 |
| Managerial Experience | 4.46 | 0.67 |

The interviewees seemed to agree that management experience is important. Participant 9.N-B, for example, stated:

> (I)t is very important that these people are familiar with operations, and it is preferable if they have managed a company. It is important so you can understand what is needed on the board and be capable of understanding . . . the operations of the company [when] you are an NC member.

The interviewee further stated that operations are similar in many companies and therefore it is crucial that NC members have managerial experience. Participants 1.N and 6.N agreed that experience was important; participant 6.N stated that she was once the only member

on her committee who had experience in managing a company: "The other two members frequently looked to me for my opinion". According to participant 5.N, it was tricky to answer who should be appointed as an NC member, but he said there should be broad experience, as well as diversity in gender, age, and experience.

Experience as a board member was also often mentioned when discussing managerial experience during the interviews. The survey results suggest that board member experience is the third most important quality (M 4.27, SD = 0.73). According to some of the interviewees, experience as a board member and management experience are vital. Participant 8.N-B stated: "If you do not have management experience or experience as a board member, you do not even know what you should ask or what you should be looking for when you are trying to create a board dynamic". Participant 9.N-B agreed that it was an advantage for NC members to have experience on a board. He said:

> In the future, I think we will have older men . . . on these committees . . . . [W]e will have someone that has experience in operations and experience as a board member so that they know how the dynamic should be . . . . These two factors are particularly important.

Participants 2.N, 5.N, 7.N, and 13.C agreed that board experience was important. Participant 2.N stated:

> I feel it is important you have someone that has been in that position, and it has definitely helped me in my work as an NC member, as you can easily put yourself in their shoes as you have been there before . . . and you learn from experience, and I feel that is more important than to have recruitment specialists.

Looking back to the survey results, the second most important requisite for NC members was corporate governance knowledge (M = 4.31, SD = 0.64). However, during the interviews, this knowledge was not mentioned as frequently. Participant 1.N said she understood why there might be a need for lawyers on the committee, especially those with corporate governance knowledge. Participant 10.B also stated that NCs should be more like Audit Committees, meaning that the committees should include professionals and perhaps experts on boards and corporate governance.

Two types of knowledge were frequently mentioned during the interviews and in comparison with the survey were considered the two least important factors out of the five. In the survey, knowledge of recruitment was considered the second least important (M = 3.64, SD = 1.04). When mentioned during the interviews, the interviewees' backgrounds influenced the focus on recruitment knowledge. For example, participants 1.N and 6.N, both recruitment specialists, stated that it might be beneficial to have someone on the committee with recruitment experience. Participant 1.N said she understood why it was good to have someone on the committee who has a strong human resources background, either as a consultant or a human resource manager. Participants 1.N and 6.N both agreed that recruitment specialists or consultants are often chosen for committees as they know the market and are involved in hiring. Some respondents felt that it could be beneficial for a committee to have someone who knows the hiring process. Participant 11.I, though not a recruitment specialist, also recognized this, as she explained that there is a more structured process when someone on the committee has human resource experience: "There is more structure as you start here . . . while other committees are just quite jolly with less structure". She further stated that it would be good to have someone on the committee who was strong in human resources and with contacts in the Icelandic business sector. Participant 4.N, on the other hand, did not think that someone from a recruitment agency should be a member of an NC, as the committee could seek assistance from or consult external recruitment agencies if needed. She added:

> By having someone from a recruitment agency, the recruitment offices are disqualifying themselves in a number of cases if they have their representative on the committee and therefore, I think NCs should only seek advice from the agencies if it is needed.

Finally, in the survey results, legal knowledge was deemed the least important out of the five options (M = 3.09, SD = 0.95) and was rated relatively lower than recruitment knowledge (M = 3.64, SD = 1.04). Although legal knowledge was not deemed as important as other knowledge and experience, some interviewees found that it can be beneficial to have someone with legal knowledge within an NC. Accordingly, interviewee 6.N explained the benefits of having a lawyer on the committee: "[T]here are certain obligations and rules that need to be met . . . so I would recommend the NC keep that in mind". She acknowledged that she was a member of a committee without a lawyer, but the NC had access to a company lawyer who worked with the committee when necessary. Participant 4.N, a lawyer, said that NCs needed to have a lawyer as a member because of the legal environment in which committees work and to help with legal interpretations. However, she maintained that a lawyer's experience and background needed to be considered as well. Participant 1.N, in her experience, had not been on an NC that had a lawyer and said this was not problematic, but, as mentioned above, she acknowledged that it could be beneficial.

One-way ANOVA analysis showed that there was a significant difference between people with legal education and individuals with business, economic, social sciences, engineering, and science education when asked about the importance of legal knowledge (F(3.138) = 4.904, $p < 0.05$). The results imply that participants with legal knowledge are more inclined to find legal knowledge of NC members to be important when selecting NC members. Significant differences were not found with respect to other questions or between other educational fields. When comparing the four different stakeholder groups and how they answered these questions in a one-way ANOVA analysis, there was only a significant difference between NC members and others in the first statement, i.e., regarding the importance of recruitment knowledge (F(3.131) = 4.029, $p < 0.05$). NC members (M = 4.10) found that recruitment knowledge was more important than those who were not NC members, investors, or board members of listed companies (M = 3.21). This implies that NC members seem to consider recruitment knowledge more important than the other group but not more important when compared to board members or shareholders. There were no significant differences with respect to any other questions between NC members, investors, board members of listed companies, and others. An independent *t*-test showed that for these statements there was not a significant statistical difference between males and females across the different groups.

### 4.2. Qualities and Composition of NC Members

There are aspects other than knowledge and experience that need to be considered when choosing NC members, and all 13 interviewees mentioned some qualities that can also be important for NC members to possess. Survey participants were also asked which qualities were important for NC members to possess on a scale from one to ten. The list can be seen in Table 6, along with how participants rated each characteristic.

**Table 6.** Which of the Following Characteristic Features are Important for NC Members to Possess?

| Characteristics | Mean | Standard Deviation |
| --- | --- | --- |
| Good morals | 9.24 | 1.56 |
| Independence in decision making | 9.03 | 1.59 |
| Capacity to stand by their conviction | 9.01 | 1.68 |
| Professional behaviour | 8.97 | 1.55 |
| Enough time and interest to carry out the committee work | 8.79 | 1.62 |
| Open-minded | 8.74 | 1.60 |
| Critical thinking | 8.71 | 1.69 |
| Communication skills | 8.25 | 1.56 |
| Analytic skills | 8.07 | 1.75 |
| Organizational skills | 7.46 | 1.67 |
| Strong network | 6.97 | 2.32 |
| Negotiation skills | 6.66 | 2.08 |

The results show that it is most important for NC members to have good morals (M = 9.24, SD = 1.56). The next two important qualities are NC members' independence in decision making (M = 9.03, SD = 1.59) and their capacity to stand by their convictions (M = 9.01, SD = 1.68). Professional behavior (M = 8.97, SD = 1.55), enough time and interest to carry out the committee work (M = 8.79, SD = 1.62), and being open-minded (M = 8.74, SD = 1.60) were ranked in fourth to sixth place and deemed to be more important than critical thinking (M = 8.71, SD = 1.69), having communication skills (M = 8.25, SD = 1.56), and analytic skills (M = 8.07, SD = 1.75). Comparing these results with the interviews, a few of these qualifications were also mentioned by the interviewees: open-mindedness, critical thinking, professionalism, communication skills, and being able to stand by their own convictions. According to participant 1.N, the following aspects were also important:

> They have to be open-minded and have critical thinking . . . you have to make sure there is professionalism and honesty, and you need to understand the situation and try to understand there are different points of view, for example between institutional investors and private investors, and you need to understand those different views.

Participant 4.N also pointed out that NC members needed to be able to communicate with people, be tolerant, and have some neutrality when judging others. She stated: "We are all human . . . we need to ... put aside personal issues and maintain professionalism". Participants 8.N-B and 13.C both agreed on the necessity of having strong individuals on the committee. Participant 13.C acknowledged: "You cannot have people that always say, 'yes' and you need to have someone that participates in the discussions and has the courage to, for example, go against the current board members".

The survey showed that only two features scored below 7 in the survey and were ranked the lowest of the twelve features, namely, negotiations skills (M = 6.66, SD, 2.08) and strong network (M = 6.97, SD = 2.32), these being considered slightly less important than organizational skills (M = 7.46, SD = 1.67).

When looking at the quality of having a strong network, which was ranked as the second least important in the survey, it is interesting to look at comments from interviewees regarding networks. Participant 5.N, for example, admitted that, although it had worked in the past to use his network and contact people to try to find board members, he would not recommend that approach today, as it was not professional. Other interviewees also focused on how important it is for NC members to have a vibrant network. Participant 13.C, for example, said: "NC members need to know their business segment so they can possibly find some suitable candidates". Interviewee 12.I also stated that it is important for a global company to have NC members who can source people from abroad. Interviewee 9.N-B said that the NC members' network is often used to find board members. Interviewee 8.N-B also expressed that the NC's network can be increased by having a diverse NC. Participant 4.N mentioned that women have smaller networks "in this world" than males, so NCs can increase the access of women to apply for a board seat when companies have an NC and board members are not appointed by using someone's network. However, she also admitted that during her NC work the committee had contacted someone using their network when they needed to find a female board candidate.

When comparing the four different stakeholder groups and how they answered the question regarding a strong network, it was found that NC members feel that this is more important than shareholders, board members, and others. Despite this result, a one-way ANOVA analysis showed that there was only a significant difference between NC members and others (F(3.130) = 2.555, $p < 0.05$). This implies that NC members (M = 8.0) found a strong network more important than those who were not NC members, investors, or board members of listed companies (M = 6.30). An independent sample *t*-test did not show a significant difference when comparing answers from males and females (t(131) = 1.240; $p > 0.05$). One-way analysis for education or age did not show a significant difference between groups regarding a strong network. For other characteristics, a one-way ANOVA analysis did not show a significant difference between the stakeholder groups for any other

statement. This implies that all groups were statistically in agreement when answering these questions.

In addition to the skills mentioned above, there are also other aspects that need to be considered when selecting NC members, according to the interviewees. The diversity of NC members can be an essential factor when appointing the NC, and it is important to create a team. Participant 10.B, for example, urged for diversity and people who can work well together, while participant 12.I recommended the following for NC members:

> [T]hose who are most likely to form a great team that suits the company at each time . . . You need to have someone that is most likely to sense the . . . credibility of the candidates and might have knowledge and experience of the human resources that are available.

Participant 7.N stated the need for diversity within a committee, including a consultant, an independent individual, and a shareholder. Participants 8.N-B and 10.B, both board members, agreed with the diversity aspect. Participant 8.N-B stated: "There needs to be a mix, someone who knows the company and the dynamic within . . . and someone that can say we are looking for something to fulfill this matrix". He further stated that it could work well to have people with different backgrounds who belong to different sectors, as they can have connections with other people than the typical recruitment offices in Iceland. He said that this can broaden the pool of potential candidates, as they can come up with different people, broadening the connections within the NC.

According to participant 12.I, it can also vary between companies and their situations who should be appointed to the committee:

> [I]t can differ between companies . . . if the company is going through a crisis then it can be useful to have a committee that has managed the crisis and can solve the problems, but on the other hand, you might have a company where external conditions are demanding, and that committee would need some other qualities.

It can also be beneficial for NC members to be familiar with the company's operations. Participant 3.N, for example, said it was a benefit to have someone on the NC who was previously an employee of the company. She feels that NC members need to know the market, the company's operations, and be someone who "is inside the world of listed companies". Participants 4.N and 10.B both felt that an NC needs someone who knows the company's operations, and 4.N said: "I would think the person in question needs to have some knowledge of the market and the company". Participant 1.N added that it was important for NC members to understand the different perspectives of different investors. Participant 12.I also specified that NC members should have a "deep understanding of the business in question".

Recognizing that a company's operations can be related to the tenure of Icelandic NC members, i.e., whether they are appointed for one or two years, most of the interviewees feel they should be appointed for two years. According to Participant 2.N, there needed to be some stability within the NC: "It is good to have insight into the company, and if you are constantly changing the committee . . . you do not know the company when you start and do not know what the company needs". She felt it was an advantage not to change the committee too frequently, as the members are therefore able to get some insight into the procedures and the company's dynamics, but she acknowledged that some renewal might be good. Participant 4.N agreed there was value in having someone on the committee who knows the process and that the NC should not be replaced at once. Participant 9.N-B stated that the NC should be appointed for a minimum of two years and that the "whole process restarts" if the committee is replaced simultaneously.

Some interviewees also feel that shareholders should be part of the NC. Participant 7.N explained:

> They are the ones that create the team of managers to take the company where they want it to be . . . [T]hey have monitored the company . . . have listened to the board and can judge it from their performance and if they want manage a change or not then this should be the platform.

Participant 11.I, on the other hand, did not feel that it is always optimal to have shareholders on the NC:

> We know of a situation where large shareholders have appointed themselves as chairman of the NC and then appointed the other two members on the committee. I have criticized this as I do not feel this is good corporate governance . . . and I do not feel we have the arm's length needed in this situation.

Despite this, she said: "We just need to take special care of how these committees are appointed".

## 5. Discussion and Conclusions

This research provides findings on what factors need to be considered when selecting candidates to take a seat on NCs—specifically, what knowledge, experience, and qualities are deemed to be important for NC members, according to different groups of stakeholders. The results indicate that diversity regarding experience and knowledge is most important when choosing NC members. This might not come as a surprise, as it is an important part of fostering sustainable or good governance practices of a board. Research has shown that more diverse NCs lead to more diverse boards (e.g., in terms of gender equality), which in turn fosters better ethical business practices [23]. NCs should therefore include members with different backgrounds, as this can provide committees with a wider pool of candidates, which again has a spillover effect on board compositions. Hence, as previously stated, research has uncovered that the diversity of NC members can affect the diversity of the board they nominate [9,11]. Still, research is limited regarding what qualities and experience need to be considered when choosing NC members. This research applies data from shareholders, board members, and NC members themselves to expand the literature on what knowledge and experience NC members should possess. The aim of this research is to understand what qualifications various stakeholders see as necessary when considering whom to choose to become NC members.

The findings suggest that managerial and board member experience are the top two criteria for NC members. According to Sjöstrand et al. [2], having an NC member who has previously been involved in the recruitment process can be beneficial during the process of choosing NC members and therefore beneficial for a company. Mans-Kemp and Viviers [11] and Sjöstrand et al. [2] also state that having headhunters or employees from recruitment agencies as NC members can broaden the committee's network. However, the findings indicate that knowledge of recruitment practices is not the most important factor for an NC member. At the same time, it still might be beneficial for NCs to include members with HR knowledge. This research shows that managerial experience, corporate governance knowledge, and experience as board members are all considered to be more important than recruitment knowledge. The findings reveal that it is also beneficial within an NC to have a member with considerable insight and understanding of a company's operations. This result is in accordance with the findings of Sjöstrand et al. [2], who state that an NC need to be familiar with company's operations.

The combined experience and knowledge of NC members can also be significant. The findings support the literature in that the networks of NC members are important. These provide knowledge of and access to a diverse range of candidates, which is essential in recruitment situations [2]. However, the survey results do not fully support Sjöstrand et al.'s [2] findings, as an NC member's network was considered the second least important feature for them to have. NC members confirmed that they sometimes use their networks to search for applicants, but they more frequently use advertisements and receive suggestions from shareholders.

Previous research shows that the diversity of NC members can affect the diversity of the board of directors they nominate [9,11]. Although our findings do not address the issue of whether boards nominated by NCs are more diverse, they do show that diversity in gender, age, and experience is deemed to be important. Previous research has also shown that NC members' independence was important [8]. To support that notion, our

findings suggest that independence in decision making was deemed to be the second most important qualification of NC members. Good morals were the more important. NCs, with their knowledge and qualifications, can evaluate whether necessary knowledge and skills reside within a board to deal with sustainability and good corporate governance and therefore support more sustainable board practices.

One of the benefits of this research is that it is supported by both quantitative and qualitative data. Additionally, the Iceland Chamber of Commerce was also involved in gathering data for the quantitative part of the survey and provided a large number of the participants' e-mails. The sample size would not have been as large if it were not for this cooperation. The involvement of the Iceland Chamber of Commerce helped with gaining further responses from the participants in the survey, as they promoted the survey and asked the participants to provide answers. Alternatively, one of the limitations of this research is that the quantitative questionnaire included questions on the Icelandic Corporate governance guidelines. This might be a disadvantage for this research, as the participants needed to answer the questions on the corporate governance guidelines before they received the questions on NCs. Consequently, the questionnaire was time-consuming to answer, which might have affected the response rate. Another limitation regarding the qualitative data used in this research is that the interpretations of the interviews might have been affected by the researcher's perspectives. This was kept in mind, both during the interviews and while analyzing the data, so as to not influence data interpretation.

There are some limitations to the interviews in that most of the interviewees were members of NCs, and only two investors were interviewed. This might have affected the results, as we found in the quantitative results that shareholders were less satisfied with NCs than NC members themselves. Having NC members participate in the survey was found to be appropriate to gain further insight into the committees and how they operate and compare different experiences (e.g., having board members on the NC). Another limitation is that the survey questionnaire used for this research was created based on the interviewees and the literature. However, we did not find a structured questionnaire to use for this survey. Additionally, most of the participants who selected the option "other" for various responses did not explain why. Therefore, it might generate better insight to only provide firm answer options instead of "other" in some of the questions.

This research shines new light on who should be appointed to NCs and provides insights to close a gap in understanding the required experience, knowledge, and qualifications of NC members. However, many questions remain concerning NCs, and further research is needed. Additional studies could focus on gaining more perspectives from shareholders, as the present research focused on NC members. Future research could also delve into the processes of NCs, the structure of their work, and how processes compare between Icelandic NCs. It might also be relevant to study the outcomes generated by NCs. This could highlight NC proposals and whether or not they are approved by shareholders, as well as the propensities of companies owned by institutional investors to approve NC proposals without changes. Ultimately, the NC plays a vital role in strengthening a board and the company's governance.

**Author Contributions:** Methodology, A.A.A.; Validation, A.A.A. and T.O.S.; Formal analysis, H.M. and T.O.S.; Investigation, T.O.S.; Resources, A.A.A. and T.O.S.; Writing—original draft, H.M.; Writing—review & editing, A.A.A.; Supervision, T.O.S.; Project administration, H.M. All authors have read and agreed to the published version of the manuscript.

**Funding:** This research received no external funding.

**Institutional Review Board Statement:** Not applicable.

**Informed Consent Statement:** Informed consent was obtained from all subjects involved in the study.

**Data Availability Statement:** Not applicable.

**Conflicts of Interest:** The authors declare no conflict of interest.

## Appendix A

| Background variables | Nomination Committee Members | Investors/Shareholders, Board Members, Others |
|---|---|---|
| | **Gender—Age—Education** | **Gender—Age—Education** |
| Individual (knowledge, abilities, skills, other) | Nomination Committee: Where, when did you take a seat in the committee and for how long have you been there?<br><br>- Board experience?<br>- Managerial experience?<br>- Experience of employee selection? | - What experience do you have with the work of an NC? |
| Way into NC/Relation to NC | How did it happen that you became a NC member? (make sure you mention to networks, CEO relationship) Can you describe your selection process? | What experience do you have with the work of an NC? |
| **Part 1 questions** | **Nomination Committee Members** | **Investors/Shareholders, Board Members, Others** |
| Role—benefit—success criteria | How would you define the role and purpose of NC?<br>What are the pros and cons of NC?<br>What are the potential benefits of the nomination committees over another approach to the election of directors?<br>What disadvantages are overcome by the involvement of NCs? | How would you define the role and purpose of NC?<br>What are the pros and cons of NC?<br>What are the potential benefits of the nomination committees over another approach to the election of directors?<br>What disadvantages are overcome by the involvement of NCs? |
| Which ideology are at play? | What arguments are against using NC? In others opinion? Or your opinion? others? | What arguments are against using NC? In others opinion? Or your opinion? others? |
| Impact of NC? | Do you feel that shareholders and the board of directors see actual benefits of the NC's work? | Do you feel that shareholders and the board of directors see actual benefits of the NC's work? |
| Independence and power of NC? | Have you experienced that shareholders or BoD try to influence NC's work or results? And the CEO? | Have you experienced that shareholders or BoD try to influence NC's work or results? And the CEO? |
| Who are members of NC? | Who should in your opinion be members of NCs? Knowledge, abilities, skills, others? Should minority shareholders have representatives on the NC? | Who should in your opinion be members of NC? Knowledge, abilities, skills, others? Should minority shareholders have representatives on the NC? |
| Independent of board and shareholders | Should an NC be composed only of people who are independent of the board and shareholders? | Should an NC be composed only of people who are independent of the board and shareholders? |
| Independent of the board vs. having a board member in NC | What opinion do you have of board member having a seat in the NC? a board member sitting in the NC? What are the pros and cons of having a board member on the NC? If there is a board member on the NC, should it be the chairman of the board or a director (pros and cons)? In your NC, which meeting does the board member attend and which not? | What opinion do you have of board member having a seat in the NC? a board member sitting in the NC? What are the pros and cons of having a board member on the NC? If there is a board member on the NC, should it be the chairman of the board or a director (pros and cons)? In your NC, which meeting does the board member attend and which not? |

| | | |
|---|---|---|
| Insight into company's need now and for the future, as well as the board's performance (P1 & P2 question) | How does the NC gain sufficient understanding/insight into the needs of the company? How does the NC gain sufficient understanding/insight into the performance of individual directors and the board as a whole? | How does the NC gain sufficient understanding/insight into the needs of the company? How does the NC gain sufficient understanding/insight into the performance of individual directors and the board as a whole? |
| Dedication vs. many NCs | What is your opinion of the same person having a seat on more than one NC? | What is your opinion of the same person having a seat on more than one NC? |
| Payment to NC members | How are you compensated for your participation on the NC? How should the payments be, in your opinion? | How should the NC be compensated, in your opinion? |
| Sub-committee of board vs. elected at AGM | Should the NC be a subcommittee under the board, or should it be directly under the AGM, why? | Should the NC be a subcommittee under the board, or should it be directly under the AGM, why? |
| Valuation of different stakeholders on NCs work | How do different stakeholders evaluate the work and results from the NC? What is the value of the NC's work for shareholders and the board? | How do different stakeholders evaluate the work and results from the NC? What is the value of the NC's work for shareholders and the board? |
| Part 2 questions | **Nomination Committee Members** | **Investors/Shareholders, Board Members, Others** |
| When does the work of NC start? | At what moment (reason) does the NC get together? | At what moment (reason) does the NC get together? |
| Structure of NC work | How often does the NC hold meetings and what is the scope? (hours worked) | How often should the NC hold meetings and what is the scope? (hours worked) |
| Work plan | Does the NC have a work schedule? (Get a copy) | Does the NC have a work schedule? Have you received work schedule from the NC or is it presented at the AGM? |
| Process | Can you describe the activities of your NC and the work process? | Can you describe the activities of NC? |
| Process: Preparation stage | Assessing the current situation, how? BoD, shareholders | Assessing the current situation, how? BoD, shareholders |
| Process: Analysis of future | Assessing the future, how? Who has a voice and how do you really see what needs the company has? | Assessing the future, how? Who has a voice and how do you really see what needs the company has? |
| Gap analysis | Gap analysis—is there a need for new board members? What KASO (knowledge, abilities, skills, other) is missing for the board in whole? Matrix analysis of the board as a team (get a copy)? What factors are evaluated and how? Who does have the final say in deciding the needs of a new board member and the description of his/her responsibilities at the board? | Gap analysis—is there a need for new board members? What KASO is missing for the board in whole? Matrix analysis of the board as a team (get a copy)? What factors are evaluated and how? Who does have the final say in deciding the needs of a new board member and the description of his/her responsibilities at the board? |
| Process: How are candidates found? | How do you look for candidates? Do you advertise for board members? Nominations (from board, CEO, shareholders . . . ?) Can small shareholders suggest candidates? From the network of NC members? Recruitment agencies? Do you have wide range of names? Do you screen the persons? | How should the NC look for candidates? Have you provided the NC with potential candidates? (advertise for board members? Nominations (from board, CEO, shareholders . . . ?) Can small shareholders suggest candidates? From the network of NC members? Recruitment agencies? Wide range of names? Screening of potential candidates? |

| | | |
|---|---|---|
| Analysis of application and processing of them | How are individuals and teams identified? What factors are considered? Do you have a matrix or job description? | Do you get info on how individuals and teams are identified? What factors are considered? |
| Interviews | Do you interview all the parties? Do you seek recommendations? | Has the NC contacted you for recommendations on potential candidates? |
| Analysis of data | How does the NC evaluate the results of the selection process? Who does make the final decision? | How does the NC evaluate the results of the selection process? |
| Board Dynamics | How does the NC assess what effects a new person will have on the morale of the board? | How does the NC assess what effects a new person will have on the morale of the board? |
| If not functioning board, what then? | If the NC feels that the board is not functional, but all board members want to continue the board, what does the NC do? | If the NC feels that the board is not functional, but all board members want to continue the board, what does the NC do? |
| Transparency in approach? | How transparent do you feel that the process/work of the NC is? | How transparent do you feel that the process/work of the NC is? |
| Naive of very formal? | How does the NC shape its approach? How rigorous are its processes? How much preparation and analysis of the approach? | How should the NC shape its approach? How rigorous are its processes? How much preparation and analysis of the approach? |
| Reasoning of choices | How does the NC substantiate/justify the nominations? | How useful is the NCs substantiation/justification for its nominations? |
| Introduction of results | How does the NC present its results? Do you nominate individuals or specific group? (Does the AGM have many nominees to choose from?) | How should the NC present its results, nominate individuals or specific group? (Should the AGM have many nominees to choose from?) |
| Are applicants withdrawing their applications? | Have applicants withdrawn their application? | Have applicants withdrawn their application? |
| Development of NC | How do you think the NCs are going to develop in the future? What are the main issues that NC face today? What are the toughest challenges an NC faces? | How do you think the NCs are going to develop in the future? What are the main issues that NC face today? What are the toughest challenges an NC faces? |

## Appendix B

Survey Questionnaire
Background variables

A.  Gender (male/female/other)
B.  Age (<30/30–39/40–49/50–59/>60)
C.  Education (Law/Economics-Business-Social Sciences/Engineering and Natural Sciences/Health Sciences/Other)
D.  Are you a member of a NC (Yes/No/No, but I have previously been a members of an NC)?
E.  Are you a board director (board member) (Yes/No)

    a.  If Yes: What type of a company are you a board director at (Listed/Not listed)
    b.  If Yes: Are you a chairman of the board (Yes/no)

F.  Are you a Shareholder or representing group of investors (Yes/No)

    ●  If Yes:
    ●  Your involvement is . . .
       ○  Institutional investor

      o      Investor with private equity
      o      Other _______

**Questions on the Corporate Governance Guidelines issues by the Chamber**

1. How frequently do you believe it is necessary to update the Corporate Governance Guidelines?

    a. Every year/2-3 years/5 years/More seldom/No opinion

2. Do you believe that certain parts of the governance guidelines need to be revised now? (Yes/No)

    a. If Yes: Shareholders and shareholders 'meetings/Board/Directors of boards/CEO/Board sub-committees/Information on corporate governance/Other

3. Do you think it's clear how to follow the governance guidelines? (Yes/No)

    a. If No: What is not clear or creates uncertainty?

4. Is there a need for more thorough directions on guidelines 'sections? (Yes/No)

    a. If Yes: Directions about shareholder meetings/about directors 'responsibilities (legal)/about sub-committees/about the relationship between the board and the CEO/on how to deal with gender quota/on the line between law and guidelines/Other.

5. Has your firm had to deviate from the guidance of the guidelines? (Yes/No)

    a. If yes, how?

6. Has it been clear how to follow "Comply or Explain" rule? (Yes/No)

    a. If yes, how?

**Questions on Corporate Governance Guidelines concerning Nomination Committees**

7. Has the company you represent established a nomination committee?

    a. Yes/No/Do not know.

8. In your opinion, are the provisions regarding NCs clear/helpful?

    b. Yes/No/Do not know.

9. Do you think more detailed provisions on NC's are necessary in the guidelines?

    a. Yes/No/Do not know.
    b. If yes than explain:
    c. Can you further explain the need for more detailed provisions on nomination committees?

10. How much do you agree or disagree with the following statement? The Governance Guidelines should discuss how to report on sustainability and non-financial information (disclosure)!

    a. Much Agree/Agree/Neutral/Disagree/Much Disagree/No opinion

11. Do you believe that the Guidelines should provide information on how to establish a strategy on gender balance at the managerial level (11)?

    a. Yes/No/Do not know.

**Questions on Nomination Committees**

12. Should NC's be governed by the board of directors (BoD) or shareholders (AGM—the annual general meeting)?

    a. Board of directors
    b. Shareholders
    c. Both
    d. I do not have an opinion.

13. Should board members have a seat on the NC?

    a. No

b. Yes, only one board member
c. Yes, the majority of NC members should be board members
d. Yes, the NC should only include board members
e. Other

14. Who should elect members for the NC?

a. Board
b. Shareholders
c. Shareholders should elect majority of NC members while the board should elect one member
d. The board should elect majority of NC members while the shareholders should elect one member
e. Other, what?

15. How do you think NC members should be paid for their work? Payment for each meeting

a. Payment for each meeting
b. Payment per hour worked
c. Monthly payment
d. Lump sum payment for each year (amount decided during AGM)
e. Other ___________

16. How strongly do you agree or disagree with the following statements? (5-point scale: strongly agree, rather agree, neither, rather disagree, strongly disagree)

a. The information provided to the NC are not classified as insider trading information
b. Information about future directors and information on who are running to become board members are considered information that is likely to have a significant impact on the market price of a company.
c. NC appointed by independent parties can decide on who should be appointed as directors without access to insider information.

17. How strongly do you agree or disagree with the following statements (7-point Likert scale: strongly agree, rather agree, agree, neither, disagree, rather disagree, strongly disagree):

a. NCs have an important role for the formation of boards of listed companies
b. The process of forming boards is more professional when NC's are operating
c. The process of board formation is more transparent when the company has an NC
d. The nomination reports from NC's are generally well-reasoned
e. I am generally satisfied with the work of NCs
f. Arrangements for appointment of board members was better before NCs were established
g. There is no need for the NC if all board members intend to continue as directors on the board

18. How strongly do you agree or disagree with the following statements on knowledge and experience of NC members? (5-point scale: strongly agree, rather agree, neither, rather disagree, strongly disagree)

a. knowledge in recruitments
b. legal knowledge
c. corporate governance knowledge
d. Experience as board member
e. managerial experience

19. Which of the following features is important for NC members to possess? (10-point scale (1 = very lightweight—10 = very important)

a. Good morals

b. Open minded
c. Organizational skills
d. Communication skills
e. Analytic skills
f. Negotiation skills
g. Critical thinking
h. Independence in decisions making
i. Capacity to stand by their conviction
j. Professional behavior
k. Enough time and interest to carry out the committee work
l. Strong network

**Questions for NC members (only for those who are currently, or have previously been, members of NCs.**

20. Are you the chairman of the nomination committee?
    - Yes/no

21. How did it come about that you were selected for the nomination committee? Mark all that applies.
    - Equity ownership
    - Personal ties
    - Family ties
    - Ties with the CEO
    - Ties with other manager of the company
    - Ties with former board member
    - Ties with board member of another company
    - Appointed by third party, e.g., pension fund
    - Trough recruitment agency
    - Trough advertisement
    - Other, what?

22. Does a board member have a seat on your NC?
    - Yes/No

23. Is your NC governed by the board or shareholders?
    - Shareholders/BoD/Do not know

24. How much do you agree/disagree with the following statements? (5-point scale)
    - I believe my NC can receive sufficient information through meetings and other means to decide on new board members or changes to the board without a board member being on the NC
    - Alternate members should be appointed for the NC in addition to the three members that are appointed.
    - I feel that shareholders try to influence the work of my NC
    - I feel that directors of the board try to influence the work of my NC
    - I feel that the CEO tries to influence the work of my NC
    - Shareholders generally ask for a meeting with the NC
    - I think the BoD is satisfied with the work of the NC
    - I think that the shareholders are satisfied with the NCs work
    - I think that the management is satisfied with the NCs work

25. When acquiring candidates for the Board of Directors, the following methods are used (always, often, sometimes, rarely, never, do not know):
    - Advertising in the media
    - Advertising on the company's website and Icelandic stock exchange
    - Shareholders make suggestions
    - Suggestions from institutional investors

- Suggestions from the board
- The CEO makes suggestion
- Suggestions from other members of the management
- Personal network of committee members
- Recruitment office hired to submit proposals of candidates
- Other, what

26. Do you interview all applicants?

- Yes/No/Only few that are believed to meet requirements

27. Do you seek/ask for recommendation for eligible candidates

- Yes/No/only if no one in the NC knows the person

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
