# Peer review of "Selecting Nomination Committee Members—Stakeholders’ Perspective"

_sustainability, doi:10.3390/su15065595_

Round 1
Reviewer 1 Report
Dear Authors,
Congratulations on the great data collection. The paper has great potential with minor revisions. Choosing the mix of qualitative and quantitative research methods gives you the possibility to have good results. But, unfortunately you have a few problems that need to be addressed:
1. In the aim of this paper you stated that ” The aim of this research is to understand what qualifications various stakeholders do see as necessary when considering whom to choose to become nomination committee members.”. Please reformulate this according to the two groups of questions in your questionnaire, in your interview and your first sentence mentioned in your conclusion.
2. It is not clear what questions you had in your interview. From the article it is not clear what the link is between the interview questions and the questionnaire.
3. In the qualitative methodology you mentioned building codes and themes. None of those were presented in results and conclusions.
4. You should add a few more recent sources in your references answering your main research objectives (for example: how did you select the variables for your questionnaire)
Thank you and I wish you good luck
Author Response
Dear reviewer,
Please find attached our response to reviewers comments and suggestions on the manuscript. Thank you very much for contributing to improvements of the manuscript.

Reviewer 2 Report
Reviewer comments on Manuscript ID sustainability-2190552
The manuscript entitled “Selecting Nomination Committee Members – Stakeholders’ 2 Perspective”, which you submitted to sustainability, has been reviewed. Interesting topic and good effort. Based on the thorough review of the manuscript, I think the paper needs to improve elements as pointed out below.
n The study is performed in the context of the Nordic Countries including Iceland. It is necessary to mention it in the title at least in the abstract.
n In the introduction, the core methodology applied in the overall research must be explained. Please provide the research methodology this study adopted based on previous research and literature reviews.
n The research design and methodology should be represented in graphical form preferably. It helps authors and audiences to stay in the end-to-end process in mind during reading.
n For the qualitative study, the in-person survey result’s reliability must be tested. Using a test method such as the Hollis reliability test for Delphi Survey, it is confirmed that the participants’ responses are consistent.
n For the quantitative study, the actual survey questionnaire should be provided to the audience for understanding questionnaire items and the structure of the survey. Also, the reliability and validity of the survey questionnaire must be tested. In current manuscript, it cannot be found whether the 5-point Likert scale questionnaire is composed and responded to as authors intended or not. To ensure the survey data reliability and validity, the factor analysis method can be useful.
n The discussion part only describing ranking of the survey items. Mean and SD values provide no more than order of the response items which can be performed by simple calculation tools. A statistical tool should be used when validating among the relationships among the variables (independent, dependent, moderating and mediating so forth).
n In discussion, new findings and insights must be mainly described. It is difficult to find any additional findings or insights compared to the previous studies. Regarding ANOVA tests, please provide specific interpretation for the results. For example, a significant difference was (not) found, and the result implies what phenomenon.
n The conclusion must focus on the paper’s practical and theoretical contribution. Now it is more like a summary of the survey results. Please consider rewriting the conclusion part emphasizing values and insights gained by the analysis results from this study. The conclusions must provide a brief description of crucial tasks performed, valuable outcomes and contributions of the research thoroughly. Partial elements are there but need modification.
n The study seems to deal with a particular environmental condition – selecting nomination committee members in Nordic countries. Please provide the logic the results of this study can be generally accepted.

Author Response

(The authors gave the same response as above.)

Reviewer 3 Report
I am grateful for the possibility to become familiar with this manuscript. In general, albeit its subject-matter and research are maybe not the most innovative and/or groundbreaking, the paper reads smoothly and, overall, has a merit.
I would just recommend to modify the abstract. An abstract should briefly, concisely and clearly summarize the gap(s), basic goals, methods, results and conclusions of the study undertaken in the article. The structure of the present abstract should follow more exactly the IMRAD scheme and highlight the key gap(s) in theory and policy, then the corresponding goals of the article (to bridge the gap(s)), and then paper's major findings. When searching in a database, the abstract is a basis for the reader’s decision to download the paper, read it and – maybe - quote it.
Author Response

(The authors gave the same response as above.)

Round 2
Reviewer 2 Report
Reasonable effort to improve the overall quality is made.
Author Response
Please find attached a word doc with replies to reviewers.
